# Study on the Microstructure, Mechanical Properties, and Erosive Wear Behavior of HVOF Sprayed Al_2_O_3_-15 wt.%TiO_2_ Coating with NiAl Interlayer on Al-Si Cast Alloy

**DOI:** 10.3390/ma13184122

**Published:** 2020-09-16

**Authors:** Marzanna Ksiazek, Lukasz Boron, Adam Tchorz

**Affiliations:** 1Department of Non-Ferrous Metals, AGH University of Science and Technology, 30 Mickiewicza Ave., 30-059 Cracow, Poland; 2Łukasiewicz Research Network-Cracow Technology Institute, 73 Zakopianska St., 30-418 Cracow, Poland; lukasz.boron@kit.lukasiewicz.gov.pl (L.B.); adam.tchorz@kit.lukasiewicz.gov.pl (A.T.)

**Keywords:** Al_2_O_3_-TiO_2_ coating, HVOF process, erosion wear, Al-Si cast alloy

## Abstract

Alumina oxide coatings are widely used in many industrial applications to improve corrosion protection, wear and erosion resistances, and thermal insulation of metallic surfaces. The paper presents study of the microstructure, mechanical, and wear properties of HVOF (high velocity oxy-fuel process) sprayed of Al_2_O_3_-15 wt.% TiO_2_ coating with the NiAl interlayer on the surface of Al-Si alloy castings. The microstructure of Al_2_O_3_-15 wt.% TiO_2_/NiAl coating was characterized by light microscopy, X-ray diffraction (XRD), scanning electron microscope (SEM), and energy dispersive X-ray spectroscopy (EDS). The analysis of the microstructure showed the formation of coating with low porosity, compact structure, good adhesion to the substrate with typical lamellar structure composed of a solid phase consisting of compounds included in the coating material and their phase variations. For analysis of the adhesion of coatings to the substrate, the scratch test was applied. An assessment of the erosive wear resistance of coatings was also carried out, confirming the significant impact of the interlayer as well as the microstructure and phase composition of the oxide coating on the wear resistance of the tested coating system. Moreover, the results were discussed in relation to the bending strength test, including cracks and delamination in the system of the Al_2_O_3_-15 wt.% TiO_2_/NiAl/Al-Si alloy as microhardness and erosion resistance of the coating. It was found that the introduction of the NiAl metallic interlayer significantly increased resistance to cracking and wear behavior in the studied system.

## 1. Introduction

Aluminum oxide-based compositions are an important group of ceramic coating materials with many advanced properties such as heat resistance, corrosion resistance, wear resistance, chemical resistance, and excellent mechanical properties such as hardness, fracture toughness, and strength. Thermally sprayed coatings based on Al_2_O_3_ have been extensively tested over the last decade, especially those containing various TiO_2_ additives [1,2,3,4,5]. The machinability, wear resistance, or conductivity of the coatings can be adjusted by changing the TiO_2_ content. Coatings, which are constructed from a combination of these oxides, have better properties than both oxides used separately, which gives a wider range of application of these coatings. The most commonly used mixtures are Al_2_O_3_ + 3 wt.% TiO_2_, Al_2_O_3_ + 13 wt.% TiO_2_, and Al_2_O_3_ + 40 wt.% TiO_2_. Among them, the best wear resistance and toughness is achieved with coatings with the addition of 13 wt.% of TiO_2_ [6]. Coatings sprayed with Al_2_O_3_-40 wt.% TiO_2_ are mainly used as thermal barriers to protect the piston bottoms of diesel and spark engines. However, in the Al_2_O_3_-40 wt.% TiO_2_ coating, β aluminum titanate Al_2_TiO_5_ (orthorhombic system) was detected and in much lower contents of Al_2_O_3_ (corundum, hexagonal system) and TiO_2_ (rutile, tetragonal system). Aluminum titanate Al_2_TiO_5_ shows a significant anisotropy of thermal expansion coefficients, which results in the formation of a network of microcracks in the cooling grain of the coating, which reduces the performance properties of the coating [7]. Moreover, as the TiO_2_ content increased, lamellar cracks and cracks perpendicular to splats increased [8].

Current applications of Al_2_O_3_-TiO_2_ coatings commonly used in tribological applications on various mechanical parts. These coatings increase the service life of these materials, making them resistant to erosion and corrosion. Moreover, oxide coatings are mainly applied to protect Al alloys, used in the production of moving components of gas turbines (blades, discs) and internal combustion engines (valves, turbo-compressor rotors), against high temperature corrosion and oxidation, as well as to minimize damages caused by wear [9,10]. Specifically, the application of Al-Si alloys as a replacement for engine blocks made of grey cast iron, in addition to positive aspects such as the reduction of engine weight, have their disadvantages reflected primarily in the inadequate resistance to wear of this material. The still unresolved problem is the unsatisfactory heat resistance of Al alloys above the temperature 150 °C; therefore, one of the ways to achieve an increase in corrosion resistance of elements made of these alloys is to cover their surface by methods of thermal spraying with ceramic protective coatings, characterized by high melting temperature, low thermal conductivity, low density, and stable structure at the operating temperature. Therefore, successfully applying these coatings on aluminum alloys for modern equipment in the power industry is still a difficult task. The thermal flame ultrasonic spraying HVOF (high velocity oxy-fuel) technique is the most suitable for the production of high-quality wear resistant coating. Compared to other thermal spray techniques, the high velocity oxy-fuel process can be used to obtain coatings with special properties: very low porosity, low surface roughness, high adhesion to the substrate, high hardness, high abrasive resistance, good wear resistance, and the ability to resist many high-temperature corrosion environments [11]. The HVOF process is by far the most versatile modern method for surface treatment in terms of economy, range of materials applied and protected, and the scope of application. This method allows for the regeneration of used parts and machine parts, which is very important in terms of saving expensive materials and energy and reducing the environmental impact of production processes.

However, oxide ceramics exhibit polymorphism [12,13], and the resulting possibility of phase transformations is a major problem at the operation of coatings made of this material. The emergence under plasma spraying conditions of the metastable variation γ-Al_2_O_3_ (with a flat-centric network and density of 3.65 g/cm^3^) results at the temperature above 900 °C in transformation of γ-Al_2_O_3_ into corundum (thermodynamically stable variation α-Al_2_O_3_ with densely packed hexagonal network and density 3.96 g/m^3^), which is accompanied by changes in the specific volume, consequently generating stresses at the interface between the ceramic coating and the metallic substrate. In addition, the significant impact on the durability of ceramic coatings in high temperature have volume changes accompanying the phase changes of the ceramic and the metal substrate, which occur during heating and cooling [14,15,16]. For aluminum alloys covered with ceramic coatings, a large shrinkage of the substrate during the cooling period causes compressive thermal stress due to their low thermal expansion coefficient. As a consequence, coating delamination at the coating/substrate interface and/or within the coating may occur [17]. Thermal stresses are already created at the initial stage of applying the ceramic coating during plasma spraying on a cold metal substrate. Then, they disappear at the temperature of plasma spraying due to the heating of the substrate and are regenerated when cooling the element with the applied coating. Too high compressive stresses in interaction with micro-cracks and other defects in the structure of ceramic coatings obtained by plasma spraying can lead to delamination at the coating/substrate interface. Inability to relax these stresses in the ceramic coating with an ionic bond causes bonds to break at the boundary of separation and their cracking and splinting from the substrate during heating and cooling of structural elements. The value of thermal stresses generated in the ceramic coating during cooling from the temperature of plasma spraying can be reduced by pre-applying to the metal substrate an interlayer with an intermediate value of the coefficient of thermal expansion [18]. Reducing mismatch of thermal expansion coefficient (CTE) between the ceramic coating and the metal substrate by introducing a plastic metallic layer between the substrate and coating can reduce excessive stresses at the ceramic/metal interface when operating under cyclic thermal changes, leading to the propagation of cracks deep into the substrate material. Although an interlayer is generally used to enhance the coating bonding and adhesion, but its effect on other properties of Al_2_O_3_-TiO_2_ coatings, particularly toughness, corrosion resistance, and wear properties, has received less attention [19,20,21]. 

The effect of the NiAl interlayer on the mechanical and wear properties of the HVOF sprayed Al_2_O_3_-15 wt.% TiO_2_ coating on the Al-Si cast alloy has been studied and characterization focused on microstructural, mechanical, and wear behavior of the Al_2_O_3_-15 wt.% TiO_2_/NiAl/Al-Si system will be made.

## 2. Materials and Methods

### 2.1. HVOF Coating Preparation

The metal substrate for applying ceramic coatings of Al_2_O_3_ + 15 wt.% TiO_2_ powder mixture by supersonic thermal spray technology HVOF (high velocity oxy-fuel) was the AK9 (AlSi9Mg) alloy, whose chemical composition was specified in Table 1, and the selected mechanical properties in Table 2. Substrate samples with dimensions of 100 × 15 × 5 mm were made of hypoeutectic alloy Al-Si with morphology and dispersion of eutectic silicon, being characteristic for a modified alloy (Figure 1).

The material used to produce ceramic coatings by technology HVOF was the powder Al_2_O_3_+15% wt.% TiO_2_—Amdry 6228 from Sultzer Metco Inc. (Westbury, NY, USA) with a grain size of −45 + 15 µm, while interlayers—the powder Metco 404NS (NiAl20) with grain size of −90 + 53 µm. The surface of substrates prior to spraying was treated by abrasive blasting with loose corundum of granulation 200 mesh. The surface roughness parameter Ra was 10.0 ± 2.6 μm. Parameters of the spraying process, used with the HV-50HVOF system kit at Pasma System S.A. (Siemianowice Slaskie, Poland), are specified in Table 3. The surface of samples was pre-applied under fixed conditions of supersonic spraying with the NiAl interlayer with a thickness of approx. 20 µm, followed by the ceramic coating Al_2_O_3_-15% wt.% TiO_2_ with a thickness of approx. 350 μm.

### 2.2. Coating Characterization

A light microscope (LM) (Zeiss, Oberkochen, Germany), a scanning electron microscope (SEM, Dual Beam Scios FEI, Eindhoven, Holand) equipped with EDS spectrometer was used to examine the microstructure and chemical composition of the coating/substrate system. The phase composition was determined by X-ray phase analysis on X’ Pert Pro Panalytical diffractometer (Malver Panalytical Ltd., Cambridge, UK) in the 20 ÷ 90° angular range with CuK radiation. The measurement of the oxide coating porosity was performed on microscopic photographs (LM) using the Aphelion 3.0 program (ADCIS French Company, Saint-Contest, France) for analysis of stereological parameters of the microstructure.

Microhardness values of the coatings were measured using the Vickers method using Hanemann microhardness meter (Carl Zeiss, Jena, Germany), mounted on a Neophot 2 microscope at a load of 100 g. The measurements of microhardness both of the coatings and interlayers were carried out on metallographic samples made on cross-sections of normal samples to their surface. The measurements of the indentation hardness were also carried out for the precise assessment of the microhardness of the coatings.

Measurements of indentation hardness (H_IT_) and Young’s modulus (E_IT_) were carried out using a multifunction measuring platform equipped with a Anton Paar microhardness meter (Anton Paar GmbH, Buchs, Switzerland). In this method, the process of pushing the intender into the material can be evaluated by measuring both force and displacement during plastic and elastic deformation. By recording the entire force application and removal cycle, hardness values corresponding to traditional hardness values as well as other material properties such as the press module can be determined. The advantage of this method is that all of the listed values can be calculated without measuring the size of the intender imprint [22]. The hardness measurement with this method was carried out for a load force of 0.05 N and a load speed of 0.10 N/min. Five measurements were taken for each sample. Nano-meter hardness measurements were made on transverse metallographic samples of coatings and the impressions made in the nan-hardness test were located within one coating lamella.

As part of the experiment, the surface roughness of coatings made by plasma spraying was measured using a confocal microscope. Three-dimensional images and their analysis allowed for a thorough understanding of the geometric structure of the studied surfaces.

For testing the thermo-physical properties of the Al-Si alloy and the coating system Al_2_O_3_-15 wt.% TiO_2_/NiAl/Al-Si alloy, a dilatometer analysis was applied—there were determined values of dimensional changes ΔL/L and linear expansion coefficients in a solid state and as a function of temperature for the Al-Si alloy of the coating system Al_2_O_3_-15 wt.% TiO_2_/NiAl/Al-Si. Samples from the Al-Si alloy, without coating and with the ceramic coating measuring Φ 3 × 30 mm were horizontally placed in Netzsch 402E dilatometer (Erich NETZSCH GmbH & Co. Holding KG, Selb, Germany) (and induction heated from ambient temperature to 565 °C at a speed of 0.5 °C/s, and then cooled at the same speed.

### 2.3. Mechanical Characterization

The strength of the coating/substrate bond was determined during a 3-point bend test on an INSTRON 8800M machine (Instron, Norwood, MA, USA) using a specially designed holder for specimens with dimensions of 100 × 15 × 5 mm. The spacing of supports was 70 mm, and the deformation rate was 1 mm/min. Three samples were used for one test. Observations of the surface of fractures after a 3-point bending test were carried out using a scanning microscope (Dual Beam Scios FEI).

The measurement of internal stresses in sprayed coatings was performed using the non-destructive X-ray method (so called sin^2^y). The examination was performed with an X-ray diffractometer, (Bruker, Billerica, MA, USA) using monochromatic radiation of a cobalt anode lamp. The internal stress was measured at 4 points on the flat surface of the specimen. Both the determination of measurement parameters and the position of diffraction lines at the assumed y angles was carried out on the basis of company programs of the APD or XRD Commander (Bruker AXSLLC, Madison, WI, USA for phase analysis, equipped with the applied research apparatus. The obtained experimental inter-planar distances *d_hkl_* and X-ray elastic constants for the tested material constituted input data for the program calculating the values of internal stresses. The internal stresses in the ceramic coating were measured with the reflex (004) and the elasticity constants: Young’s modulus 82 GPa and Poisson’s ratio 0.25.

### 2.4. Scratch Test

The scratch bond strength tests on the coating were carried out on a multifunction measuring platform (Micro-Combi Tester, Anton Paar GmbH, Buchs, Switzerland) equipped with Anton Paar scratch test heads according to the standard [23]. The tests were carried out on the cross-sectioned samples embedded in resin and then polished in a standard way as metallographic samples. The scratch test is performer under constant load and the indenter moves from the substrate through the coating into the resin in which the sample is embedded. The following test parameters have been used to produce a scratch on each specimen: indenter (stylus) type Rockwell diamond; stylus radius 100 µm; constant normal loads of 5, 10, and 20 N; scratch length 1.2 mm and scratching speed of 1.2 mm/minute. After the examination, the geometric values of the resulting one-shaped fracture were also measured: length of cone Lx, with Ly and the angle of cone (image of the cone fracture area was taken directly after the scratch under a light microscope, shown in Figure 2). Among the Lx, Ly, and cone angle values, the projected cone area, Acn = Lx.Ly (Figure 2) was selected as the most characteristic factor because only Acn showed a monotonic dependence on the scratch load.

### 2.5. Abrasion Wear Resistance

Tests of abrasion resistance in the abrasive suspension of Al-Si and Al_2_O_3_-15 wt.% TiO_2_/NiAl/Al-Si samples were carried out on the device for testing abrasion resistance of coatings and constructional materials in an abrasive suspension. An abrasion test was carried out in an aqueous suspension of Cr_3_C_2_ powder (of average grain size less than 0.1 mm) at ambient temperature, with the following parameters: test time 3600 s, rotation speed 300 L/min, applied load 50 N. The friction node consisted of a stationary plate made of the material to be tested and a steel ball rotating at a set speed. The plate was pressed against the ball with a given force. The abrasive suspension was fed to the frictional contact zone. The test course was recorded using a computer with specialized software. The software enabled real-time calculation of load, path, rotation speed, temperature of the friction pair, depth of wear and rate of wear of both friction elements. The worn surfaces were observed using a scanning microscope.

## 3. Results and Discussion

### 3.1. Microstructural Properties

Al_2_O_3_-15 wt.% TiO_2_ ceramic coating applied by the HVOF method, after initial application to the interlayer surface NiAl by the same method, has a typical lamellar structure, without cracks and with good adhesion to the substrate (Figure 3). The ceramic coating has a compact structure with a porosity of 4.8% ± 0.2. This a low porosity value due to the high impact velocity of the coating particles, which causes high density and high cohesive strength of individual splats [3,5]. An average roughness parameter of the ceramic coating surface Ra is 5.13 μm. Measurements of the coating micro-hardness show a significant dispersion, which is related to their microstructure, and basically to an optimal ratio of bi-modal coating structure (fully: partially melted regions) [16,17]. It was found that the partially melted zone showed higher hardness and elastic modulus than the fully melted zone [20]. In this study, the Al_2_O_3_-rich areas had an average hardness of 867HV0.1, while the TiO_2_-rich areas had an average hardness of 1082HV0.1. 

The ceramic coating Al_2_O_3_-15 wt.% TiO_2_ has the structure of a multiphase solid solution Al_2_O_3_-TiO_2_. Rapid cooling of the lamellae during spraying promotes the formation of metastable phases, supersaturated solutions, and a polycrystalline structure with a much higher degree of dispersion than the conventional materials. The results of X-ray microanalysis using EDS (Figure 4) indicate that the coating contains areas rich in Al and Ti. The cross sections of the coatings reveal the areas enriched in aluminum and titanium. The strips of the titanium phase are clearly visible in the aluminum matrix. The light gray regions in the coatings are attributed to the TiO_2_ phase while the dark gray to Al_2_O_3_.

The NiAl interlayer made by the HVOF method from NiAl20 powder on a substrate of the alloy AlSi9Mg (AK9) has a complex phase composition. The heterogeneity of the chemical composition, in particular, the distribution of Al and Ni in the NiAl interlayer, was confirmed by the method of X-ray micro-analysis using the EDS spectrometer in the cross-section of samples (Figure 5). The measuring points marked on this drawing from 1 to 2 with a high concentration of Ni and Al, identified on the basis of the Ni-Al phase equilibrium system [24], correspond to the solid solution γ and β on the basis of the properly ordered phase Ni_3_Al and NiAl. The average micro-hardness of the interlayer was 316HV0.1. While the average micro-hardness of the metal substrate in various micro-areas (there are dendrites of α solid solution and (α + Si) eutectic in inter-dendritic spaces) was 32HV0.1.

Phase composition tests of coatings revealed the presence (Figure 6): TiO_2_ (rutile, tetragonal system), smaller content of α-Al_2_O_3_ (corundum, hexagonal system), and high content of the meta-stable γ-Al_2_O_3_ regular variation. This phase may be formed as a result of rapid cooling of Al_2_O_3_ molten particles. The tetragonal nature of γ-Al_2_O_3_ indicates that this phase is much weaker than the α-Al_2_O_3_ hexagonal phase. The presence of α phase in the composition of the coating has a positive effect on the hardness and Young’s modulus of the coating, while the presence of TiO_2_ makes Al_2_O_3_ coating become tighter and more resistant to brittle cracking [5,19]. The high degree of mixing of Al_2_O_3_ and TiO_2_ in the mixture probably lowers its melting point (addition of TiO_2_ to Al_2_O_3_), causing a decrease in particle viscosity (at least on the surface) and improving the interlamellar contact at impact during HVOF spraying [13,14]. The lower viscosity would provide the splats with better wetting and interlamellar strength, thus increasing the strength of the coating. Al_2_TiO_5_ aluminum titanate was not found (this compound is formed as a result of the reaction Al_2_O_3_ z TiO_2_). A high temperature gradient and short time of solidification limit the possibility of reaction in the HVOF process. Additionally, the volume fractions of individual phases in the tested coating were determined (Table 4). The content of α-Al_2_O_3_ was 23 wt.%, and the content of phase γ-Al_2_O_3_ and TiO_2_ was respectively 72 wt.% and 5 wt.%.

As part of the testing of mechanical properties of the analyzed coating system, there were also made measurements of nano-hardness and Young’s modulus. The hardness of the sprayed ceramic coating depends on its phase composition (in particular the amount of phase γ, which is intrinsically lower hardness than α) and on the porosity [14,15]. The hardness gradually decreased from 15.51 GPa in monolithic Al_2_O_3_ to 12.89 GPa in Al_2_O_3_/TiO_2_ composite coatings. This effect is related to the presence of the TiO_2_ phase with a lower hardness than the α-Al_2_O_3_ phase. Moreover, the coating Al_2_O_3_-15 wt.% TiO_2_ is characterized by the Young’s modulus in the range of 192.64 GPa (in monolithic Al_2_O_3_) to 196.63 GPa (in Al_2_O_3_/TiO_2_ composite coatings). The elastic modulus is usually related to stiffness: the higher the value of Young’s modulus, the stiffer the material. As TiO_2_ has a lower mechanical strength than Al_2_O_3_, the lamellar zones rich in TiO_2_ are likely to show lower interlayer strength, thus reducing the strength of the coating [3]. It should be noticed that for the tested HVOF sprayed coating, higher values of microhardness, elastic modulus (by Oliver–Pharr formula) were obtained than for conventional plasma sprayed Al_2_O_3_-13wt. % TiO_2_ (HV = 8.18 ± 1.29 GPa, E = 141 ± 7GPa) [25].

The tests carried out showed that Al_2_O_3_-15 wt.% TiO_2_ ceramic coatings applied to the substrate of the Al-Si alloy by the HVOF method have satisfactory structural stability. Dilatometric tests have shown that the Al-Si alloy is characterized by almost a linear relationship of the relative elongation to the temperature of about 560 °C (Figure 7). After exceeding this temperature, the linear relationship of relative elongation changes into the parabolic. This is probably related to the dissolution of silicon and magnesium in α solid solution. A similar course of dilatometric curve was observed for the Al-Si alloy sample with the applied coating system. However, the application of ceramic coating with the NiAl interlayer causes a smaller increase in the relative elongation of the sample as a function of temperature.

It should be noted that also in this temperature, the course of dependence of the mean linear expansion coefficient on the temperature (defined as: β(T − T_0_) = ΔL/L(T − T_0_), where T_0_ = 20 °C), above 560 °C is different for the coating system and for the substrate (Figure 8). The change in the effect of relative elongation at 560 °C can be caused by both changes in the structure of the AlSi9Mg alloy and a partial polymorphic transformation of Al_2_O_3_ and TiO_2_ [9]. In addition, the Al_2_O_3_-15 wt.% TiO_2_/NiAl/Al-Si system shows a thermal expansion coefficient lower by approx. 20% as compared to the Al-Si alloy substrate. The average linear expansion coefficient β(T) of both the Al-Si alloy substrate and Al_2_O_3_-15 wt.% TiO_2_/NiAl/Al-Si coating system shows variability in the range of 100–250 °C, while above 250 °C, the coefficient dependence of temperature is almost stable. The presence of pores in the coating strongly affects its properties, causing a decrease in value of Young’s modulus and the heat transfer coefficient.

### 3.2. Mechanical and Wear Behavior 

Figure 9 shows a comparison of bending test results for the Al-Si alloy substrate and the system Al_2_O_3_-TiO_2_/NiAl/Al-Si in the relation of bending stress–deflection. Values of maximum bending stresses for the Al-Si alloy substrate and Al_2_O_3_-TiO_2_/NiAl/Al-Si system are 220 ± 8 MPa and 270 ± 11 MPa, respectively. In tested systems, the bending curves are of parabolic character. However, for a substrate without a coating, on the bending curve, there is a longer range of the deflection path during which the stress slightly increases and then decreases. The value of the deflection, followed by a decrease in stress leading to the sample destruction, is 2 mm. While, for the Al_2_O_3_-15 wt.% TiO_2_/NiAl/Al-Si coating system, there is not such a long range of deflection path. Comparing the obtained curves, it can be stated that for the Al_2_O_3_-15 wt.%TiO_2_/NiAl/Al-Si system, there is an increase in bending force parameters, and the deflection is shortened to 1.7 mm. It is worth noting that the Al_2_O_3_-15 wt.%TiO_2_/NiAl/Al-Si coating system is less plastic, which consequently limits the dissipation of plastic deformation energy, and the intensively increasing load causes crack propagation and small range of deflection. It is worth noting that the Al_2_O_3_-15 wt.%TiO_2_/NiAl/Al-Si coating system is less ductile, which in turn reduces the dispersion of the energy of plastic deformation, and the intense increase in load causes crack propagation and a small range of deflection.

Fractographic tests of fracture surfaces indicate that Al_2_O_3_-15 wt.%TiO_2_ coating has the grain structure close to equal-axed morphology. Cracking of these coatings occurs according to the transcrystalline-intercrystalline mechanism (Figure 10). The fine-grained structure and the developed fracture surface of the coating indicate that it has good resistance to cracking. The crack runs in the region of the coating/interlayer interface. 

Results of the calculated main stresses (σ_1_, σ_2_) produced in the Al_2_O_3_-15 wt.%TiO_2_/NiAl/Al-Si coating system and the orientation of the main stress σ_1_ of the tested sample are summarized in Table 5. In the tested Al_2_O_3_-15 wt.%TiO_2_/NiAl/Al-Si coating system, tensile stresses occur. During the HVOF process, stresses are generated in the coating as a consequence of the rapid crystallization of molten drops of powder material and during cooling as a result of differences in thermal expansion coefficients. The resulting coatings after the HVOF spraying process have no strongly defected microstructure, which confirms low deformation of the material and the formation of tensile stresses. The value of tensile stresses and their slight concentration in the area of the *coating/substrate* interface do not cause cracks or delamination there, as a result of which the mechanical durability of the coating is not reduced. The natural stresses in the sprayed coatings are the sum of stresses resulting from cooling of liquid droplets of the coating material and the stresses during cooling of the created coating, the interlayer and the substrate as a whole [18,19]. Although melting temperature and coefficient of thermal expansion of these materials vary considerably (which, as a consequence, may worsen the adhesion of ceramic coating to the metal), the lack of delamination at the *coating/substrate* interface indicates a low level of thermal stresses associated with different values of thermal expansion coefficients of the Al-Si alloy (25 × 10^−6^/K) and ceramic coating (7.8 × 10^−6^/K), which ensured the introduction of the NiAl interlayer (19 × 10^−6^/K). It was found by Chuanxian et al. [26] that the Al_2_O_3_-TiO_2_ coating had a higher thermal diffusivity than Al_2_O_3_. Therefore, the thermal stress in the former due to local heat build up would be less, and chips would also be reduced, giving higher wear resistance and reduced friction coefficient.

The results of the scratch test made on the cross-section of the Al_2_O_3_-15 wt.%TiO_2_/NiAl/Al-Si coating system at a constant load of 5, 10, and 20 N are specified in Table 6. The projected cone was increasing with an increase of the scratch load (Figure 11). For each load of the scratch, a cone-shaped crack occurs inside the ceramic coating, which indicates cohesive damage in the coating/interlayer/substrate system. However, at the maximum load (20 N), larger cracks appear around the scratch in the area of the cone (Figure 12), leading even to the destruction of coating. Such wearing process of the layer indicates it has very good adhesion to the substrate.

The results of abrasion resistance tests in wet suspension conditions for the Al_2_O_3_-15 wt.% TiO_2_/NiAl/Al-Si coating system and Al-Si alloy substrate presented on the drawings (Figure 13) indicate that the coating system shows significantly less wear as compared with the Al-Si alloy (expressed by the depth of wear). In addition, for the coated substrate, lower erosion intensity was also found. The groove for the coating system is 25% smaller than for the substrate. The morphology of the wear area of the AlSiMg9 alloy substrate and Al_2_O_3_-15 wt.% TiO_2_/NiAl/Al-Si coating system is shown on the drawing. Cracks and micro-pores can be observed on the worn surface of the coating system. The mechanism responsible for the degradation of coating under conditions of erosive wear is an abrasive wear (cracks and chipping), which in turn indicates a brittle nature of the wear [27,28,29,30].

In addition, after the erosion test, the surface of samples was analyzed on an SEM scanning electron microscope (Figure 14). Tests carried out on the chemical composition of EDS on the cross-section of wear trace confirmed the presence of chromium on the eroded sample. However, the phenomenon of erodent deposition on the surface of tested materials was identified during erosion tests only for the AlSiMg9 alloy, while in the case of ceramic coating, the phenomenon of erodent deposition was observed only after the coating and interlayer were punctured to the substrate material. The reason of this phenomenon is the erosion of Al_2_O_3_-15 wt.%TiO_2_ hard coating and the exposure of the Al-Si substrate material, which is characterized by high plasticity compared to the ceramic coating. Then takes place the deposition process of erodent particles in the base material at the bottom of the crater in the place where the coating and interlayer were punctured. As a result of the impact of abrasive particles on the ceramic coating, defects occur in the form of cracks and chipping of the coating material. The increase of erosive wear of the coating is demonstrated by the local chipping of coating and Cr_2_O_3_ erodent particles emerging on the crater surface.

The high resistance of the ceramic coating (Al_2_O_3_-TiO_2_) is influenced not only by high mechanical durability of Al_2_O_3_ particles strongly connected to the TiO_2_ phase and by low porosity (pores are channels enabling penetration of the coating structure by particles of the abrasive suspension), but also by strong connection of the coating with the NiAl metallic interlayer. As observed by other authors [25,29], the most important factor influencing the better antiwear properties of Al_2_O_3_-13 wt.% TiO_2_ coatings is resistance to crack propagation or toughness.

## 4. Conclusions

Based on tests carried out and analysis of the results, the following conclusions were made:The ceramic coating Al_2_O_3_-15 wt.% TiO_2_ produced by the HVOF method on the pre-applied by the same method, the NiAl interlayer has a typical lamellar structure composed of a solid phase consisting of compounds included in the coating material and their phase variations and pores. The microstructure of ceramic coating confirmed by X-ray analysis of the coating, which showed the presence of α-Al_2_O_3_, high content of the meta-stable γ-Al_2_O_3_ and TiO_2_ phases. In addition, the coating is characterized by low porosity (4.8% ± 0.2), compact structure (without intra-lamellar micro-cracks), good adhesion to the substrate, low level of residual stresses (σ_1_ = 256 ± 36MPa), high hardness (in the range of 867–1082 HV0.1), and satisfactory structural stability.The structure of the Al_2_O_3_-15 wt.% TiO_2_/NiAl/Al-Si coating system provides good resistance to cracking. Destruction occurs in the area of the coating/interlayer interface. In the coating system, the increase of crack resistance takes place by inhibiting or deflecting the crack in the NiAl interlayer. The introduction of the NiAl metallic interlayer with high plasticity during the spraying process prevents the plastic deformation of a substrate and reduces the value of residual stresses.Producing Al_2_O_3_-15 wt.% TiO_2_/NiAl coating improves the erosive wear resistance of the AlSi9Mg alloy. The NiAl interlayer led to an improvement in the mechanical properties of the coating surface, which effectively reduced the intensity of erosive wear. The adhesion of Al_2_O_3_-15 wt.% TiO_2_/NiAl coating to the substrate is good, exceeding the load of 20 N in the crack test, initiates cracking of cohesive character. The reason of the increased wear resistance of the ceramic coating is high mechanical durability of Al_2_O_3_ particles strongly connected to the TiO_2_ phase, high strength of inter-lamella, high hardness and Young’s modulus, low porosity and strong connection of the coating with the NiAl metallic interlayer.

## Figures and Tables

**Figure 1 materials-13-04122-f001:**
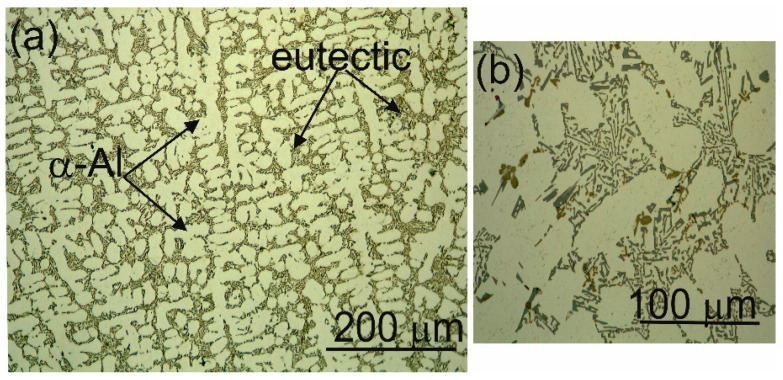
Optical micrographs showing the as-cast microstructure of the Al-Si alloy: (**a**) dendritic structure surrounded by eutectic silicon particles, and (**b**) higher magnification image showing the equiaxed coarse eutectic cell appears in which silicon grows in a modification manner.

**Figure 2 materials-13-04122-f002:**
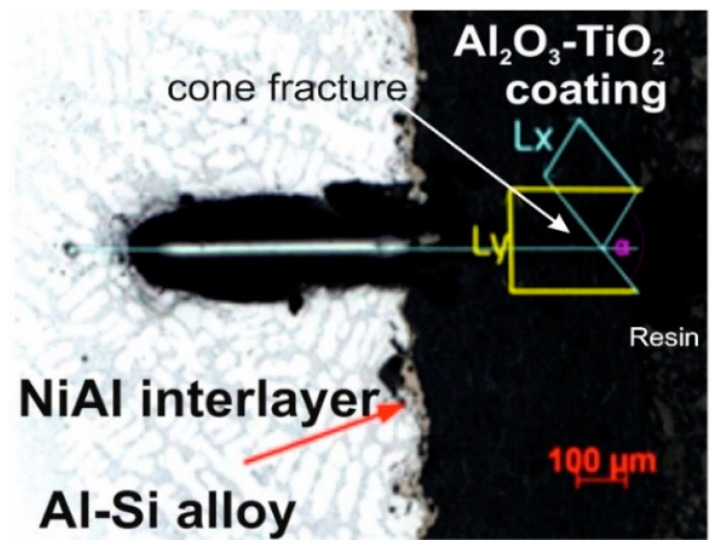
Schematic of constant load scratch on the cross sectioned sample.

**Figure 3 materials-13-04122-f003:**
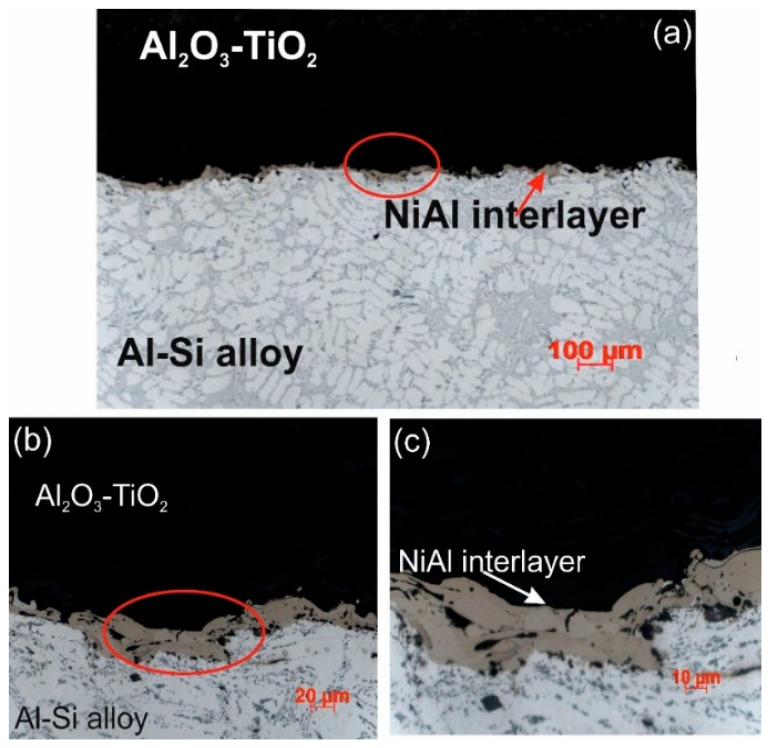
Microstructure of HVOF sprayed Al_2_O_3_-15 wt.% TiO_2_ coating with the NiAl interlayer on the Al-Si cast alloy: (**a**) LM image; (**b**,**c**) magnified area selected in Figure 3a.

**Figure 4 materials-13-04122-f004:**
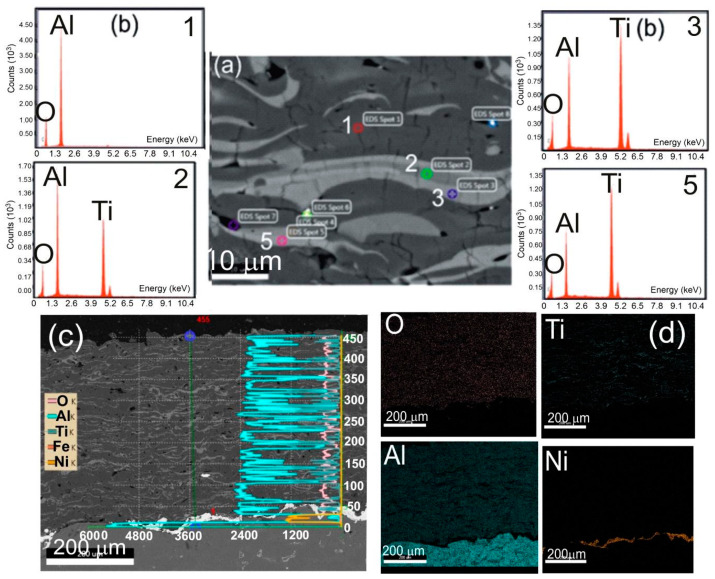
Scanning micrographs of HVOF sprayed Al_2_O_3_-15 wt.% TiO_2_ coating with the NiAl interlayer on the Al-Si cast alloy: (**a**) SEM image; with (**b**) EDS spectra taken from the marked points: 1, 2, 3, and 5; (**c**) linear distribution of concentrations of O, Ti, Al, Fe, and Ni; and (**d**) map of distribution of concentrations of O, Ti, Al, and Ni taken from the region of the *coating-substrate* interface.

**Figure 5 materials-13-04122-f005:**
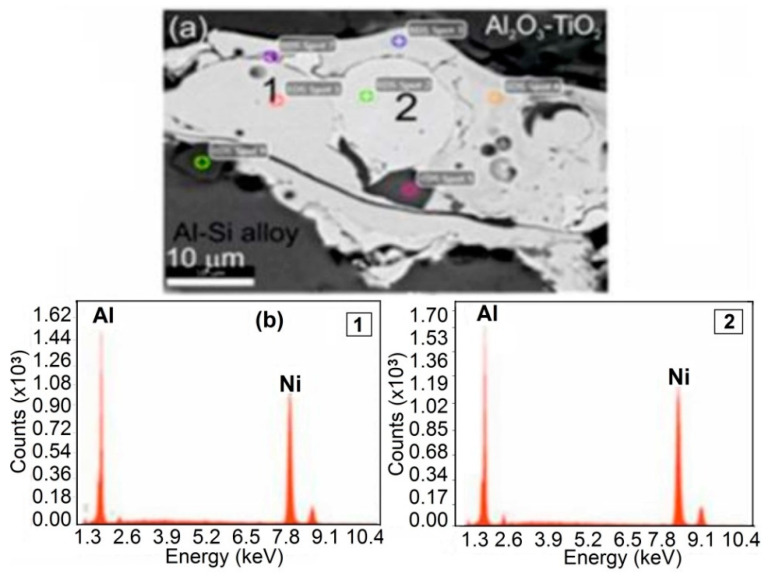
Scanning micrographs of HVOF sprayed Al_2_O_3_-15 wt.% TiO_2_ coating with the NiAl interlayer on the Al-Si cast alloy: (**a**) SEM image of the NiAl interlayer with, (**b**) EDS spectra taken from the marked points: 1 and 2.

**Figure 6 materials-13-04122-f006:**
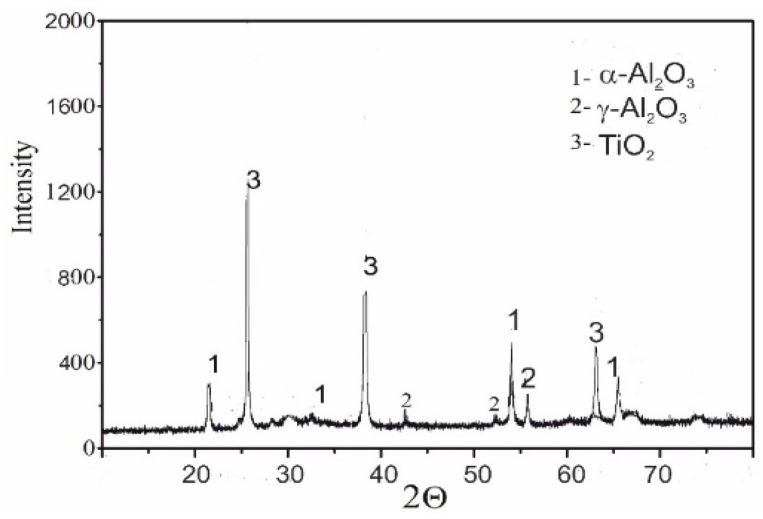
X-ray diffraction pattern of the HVOF sprayed Al_2_O_3_-15 wt.% TiO_2_ coating on the Al-Si cast alloy.

**Figure 7 materials-13-04122-f007:**
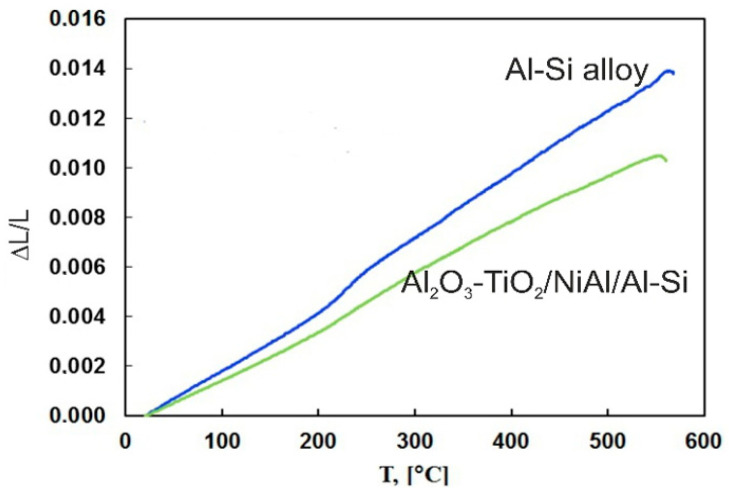
Comparison of changes in linear expansion for the Al-Si alloy and the Al_2_O_3_-15 wt.%TiO_2_/NiAl/Al-Si coating system plotted in function of temperature.

**Figure 8 materials-13-04122-f008:**
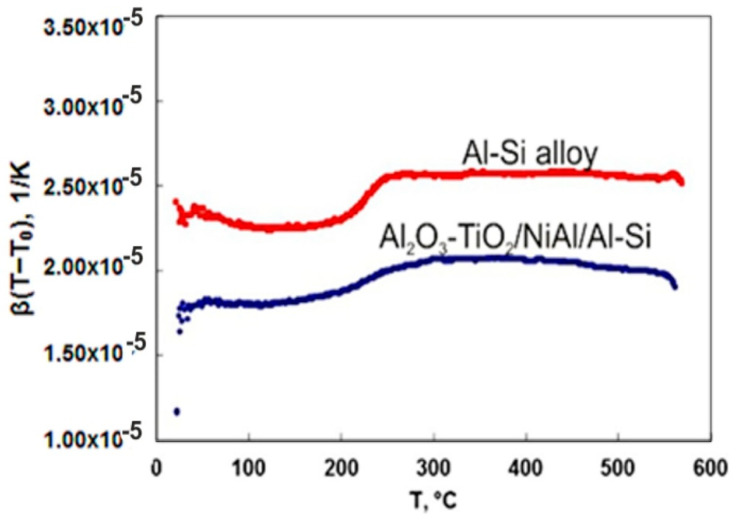
Comparison of changes in linear expansion coefficient for the Al-Si alloy and the Al_2_O_3_-15 wt.%TiO_2_/NiAl/Al-Si coating system plotted in function of temperature.

**Figure 9 materials-13-04122-f009:**
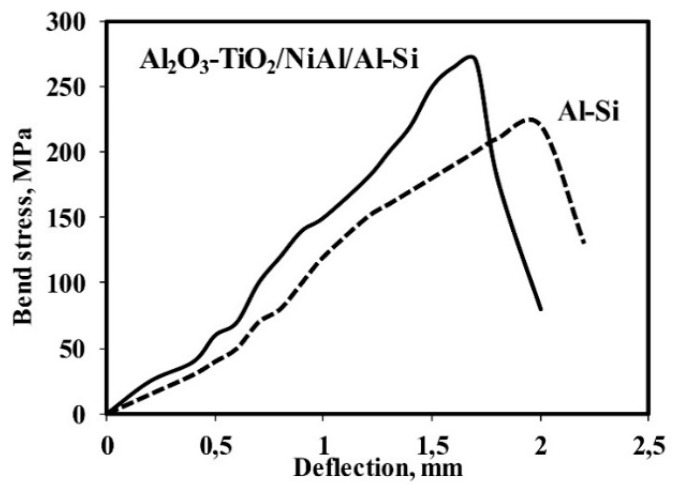
Bend test curves recorded for: the Al_2_O_3_-15 wt.%TiO_2_/NiAl/Al-Si coating system and Al-Si cast alloy.

**Figure 10 materials-13-04122-f010:**
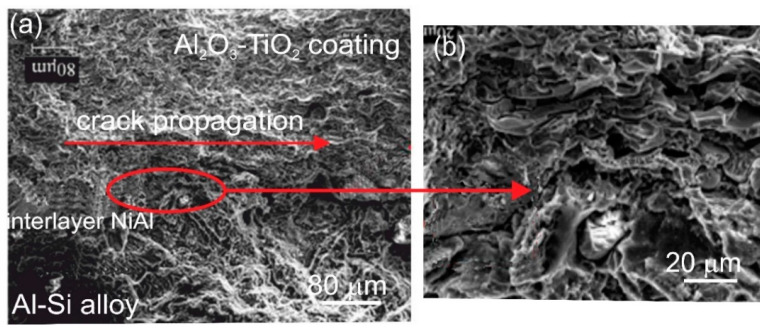
(**a**) Scanning micrographs of the fracture surface of the HVOF sprayed Al_2_O_3_-15 wt.% TiO_2_ coating with the NiAl interlayer on the Al-Si cast alloy after bend test, (**b**) magnified area selected in Figure 10a.

**Figure 11 materials-13-04122-f011:**
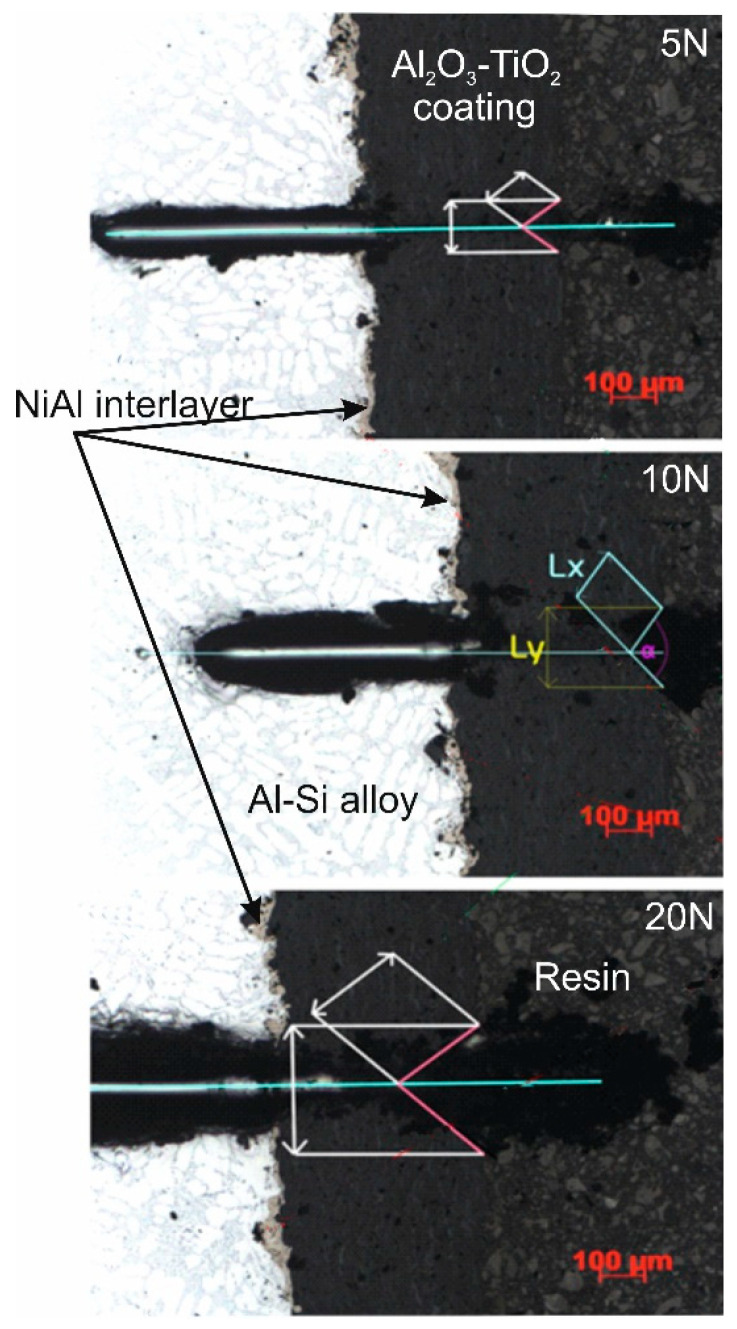
Cone-shaped crack occurring during the scratch bond test for the Al_2_O_3_-15 wt.%TiO_2_/NiAl/Al-Si coating system at scratch load of 5, 10, and 20 N.

**Figure 12 materials-13-04122-f012:**
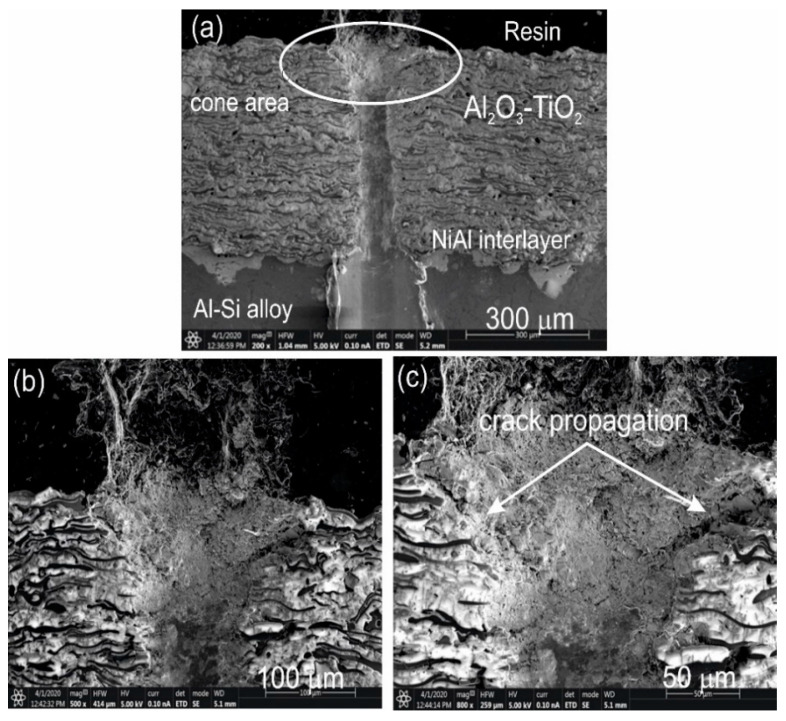
(**a**) SEM micrograph of the cross-sectional area of the Al_2_O_3_-15 wt.% TiO_2_/NiAl/Al-Si coating system cracked after the scratch test, (**b**,**c**) magnified area selected in Figure 12a.

**Figure 13 materials-13-04122-f013:**
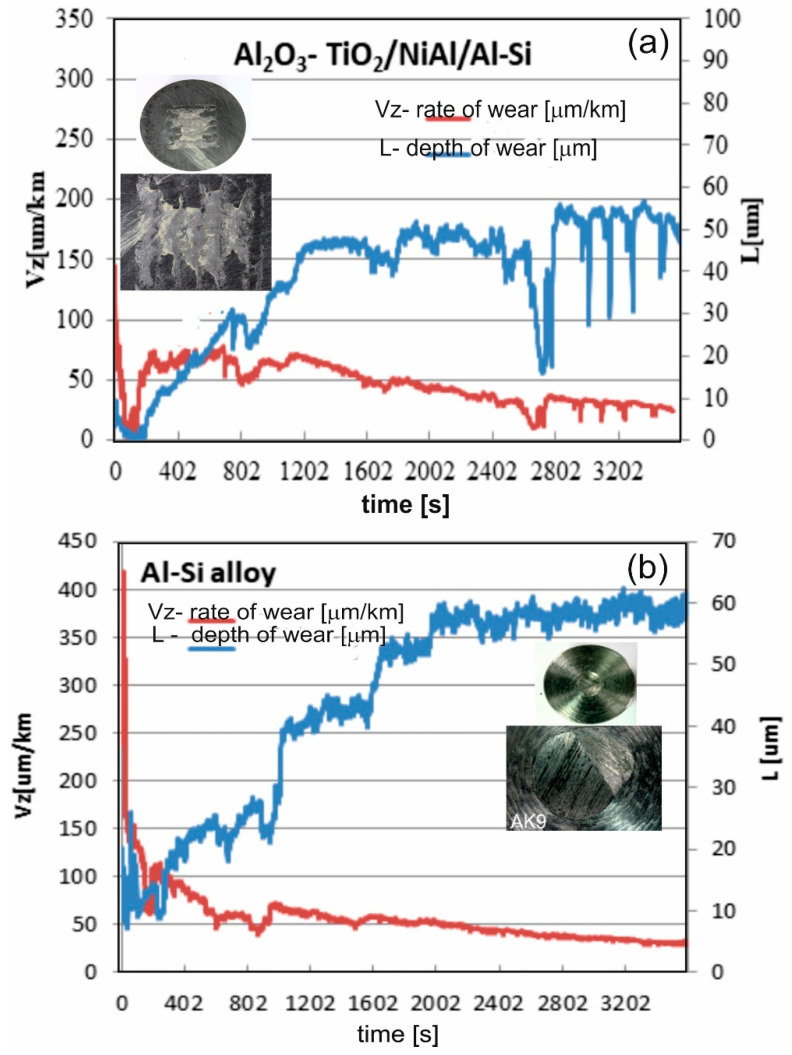
The results of the erosion resistance tests of: (**a**) the Al_2_O_3_-15 wt.%TiO_2_/NiAl/Al-Si coating system, (**b**) Al-Si cast alloy.

**Figure 14 materials-13-04122-f014:**
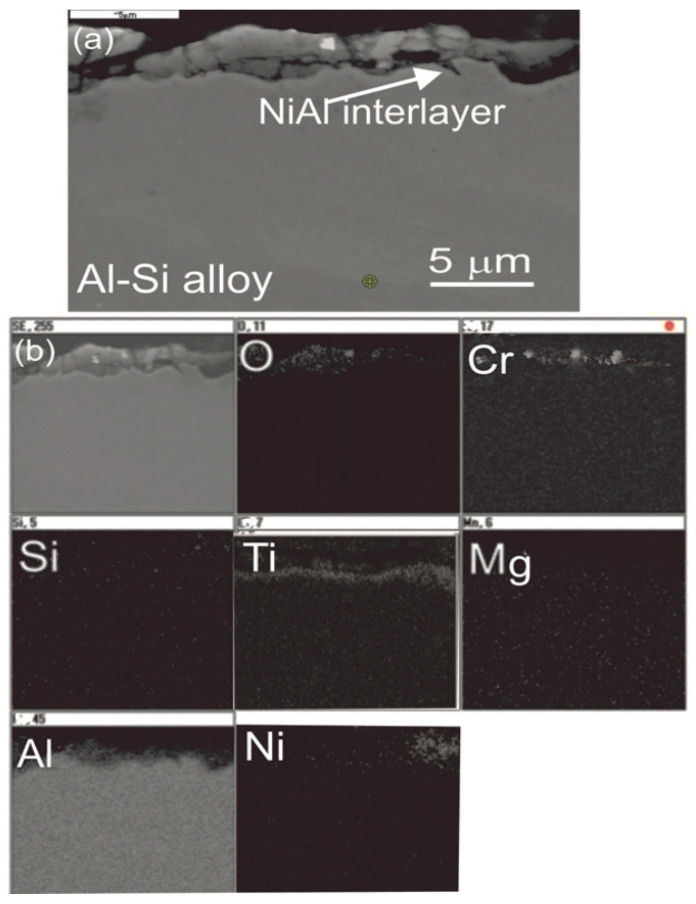
(**a**) SEM micrograph of the cross-section of the Al_2_O_3_-15 wt.% TiO_2_/NiAl/Al-Si coating system after the erosion test, and (**b**) map of distribution of concentrations of O, Cr, Si, Ti, Mg, Al, and Ni taken from the region of the interface.

**Table 1 materials-13-04122-t001:** Chemical composition of AlSi9Mg, wt.%.

Type of Al-Si Alloy	Al	Si	Mg	Mn	Cu	Zn	Ti	Fe
**AK9(AlSi9Mg)**	Bal.	10.5	0.3	0.35	0.3	0.2	0.15	0.8

**Table 2 materials-13-04122-t002:** Mechanical properties of AlSi9Mg.

Tensile Strength, MPa	Conventional Yield Point, MPa	Elongation, %	Hardness, HV
250	120	7	75

**Table 3 materials-13-04122-t003:** High Velocity Oxy-Fuel (HVOF) spraying parameters.

Gun Movement Speed, mm/s	Oxygen, L/min	Kerosene, L/h	Powder Feed Rate g/min	Powder Feed Gas, L/min	Spray Distance, mm
583	944	25.5	92	nitrogen, 9.5	370

**Table 4 materials-13-04122-t004:** Detailed results of XRD.

Composition	Weight Percentage of Phase Composition, %
α-Al_2_O_3_	23
γ-Al_2_O_3_	72
TiO_2_	5

**Table 5 materials-13-04122-t005:** Results of analysis of residual stresses in the Al_2_O_3_-15 wt.% TiO_2_/NiAl coating.

Description	Al_2_O_3_-TiO_2_/NiAl
Internal stress σ_1_ [MPa]	256 ± 36
Internal stress σ_2_ [MPa]	326 ± 40
Orientation of the main stress σ_1_ (clockwise from the direction marked on the sample)	40° ± 30°

**Table 6 materials-13-04122-t006:** Averaged scratch bond test results of the Al_2_O_3_-15 wt.% TiO_2_/NiAl/Al-Si coating system.

Load [N]	Lx [µm]	Ly [µm]	Acn × 10^−3^ [mm^2^]
5	64.12	94.84	6.08
10	133.85	204.17	27.33
20	241.64	302.06	72.99

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
