# Peer review of "Study on the Microstructure, Mechanical Properties, and Erosive Wear Behavior of HVOF Sprayed Al2O3-15 wt.%TiO2 Coating with NiAl Interlayer on Al-Si Cast Alloy"

_materials, 2020, doi:10.3390/ma13184122_

Round 1

Reviewer 1 Report

Potentially, the article presents an interesting characterization of the particular Al2O3-15wt.%TiO2/NiAl/Al-Si system, however text is poorly edited. The writing is often difficult to read, with very long sentences, repetitions and an abuse of passive forms which hinder the comprehension of the text. A lot of sentence construction errors are also present, together with some uncorrect word choices. An important revision from a native english speaker is needed before resubmission.

Introduction is not completely satisfactory. It could be enhanced by better adressing the importance of the system in the technological framework (by better citing other similar systems). A focus should be put on the role of the intermediate NiAl coating, which seems (from the text) the real novelty point of the article (in light of the performed characterizations, it appears that is the prime responsible for the enhanced mechanical properties of the system). A paragraph on the role of the intermediate coating is present in the introduction, but is generic. Also, article has few references, some more should be added.

Both "materals and methods" and "Result and discussion" are packed with informations on the used techniques which sometimes seem redundant. After the language check, i suggest to divide them in different paragraphs for the sake of clarity.

Sometimes the article reports "good performances" or "better performances" of the system, but a comparison with already-characterized systems is not present. This can be found expecially in the conclusions, and in the "Result and discussion" parts. Please add some values for comparison, or references for similar systems.

Images 4 and 14 are low quality.

Author Response

Thank you very much for the valuable comments contained in the review.

The introduction has been corrected as suggested by the reviewer (the role of NiAl interlayer  for system functionalisation was emphasized, properties were compared to similar oxide  ceramic systems).

The subsections are listed in the chapters: Materials and Methods and Results and discussion

In the results and discussion chapter, the properties of the tested system were compared, mainly with those of Al2O3-13wt.% TiO2.

The English language was improved.

Figures 4 and 14 were corrected and new references were introduced.

Please find in attachment file changes in red.

Reviewer 2 Report

The microstructure, mechanical properties, and erosive wear behavior of HVOF sprayed Al2O3-15wt% TiO2 coating with NiAl interlayer on Al-Si cast alloy was analyzed in this study. However, the authors didn’t explain the difference or improvement of the new coating method. Besides, the test lacks controlled trials. The authors should explain the innovative point of this study and the benefit to the industry.

  1. The authors should add a brief conclusion in the abstract.
  2. As the authors introduced, Alumina coatings are widely used to improve the physical properties of metallic surfaces. How did others deal with cast alloy? Besides the HVOF, Al2O3, and TiO2, authors should add more literature reviews of other coating studies and reduce the details of HVOF methods.
  3. Once the authors summarized and compared other coating studies, the research gap of previous coating methods should be given before the aim of this study. The authors should also explain why they choose the Al2O3 +15wt% TiO2 coating method.
  4. Why 15wt% TiO2. Is there any special reason or just a random selection?
  5. The main issue for this manuscript is the lack of enough controlled trial. The authors only show the improvement between the coated material and the original alloy. What’s the difference between the studied coating method and previous methods? Compared to the previous coating method, which physical properties improved more by using the new method?
  6. If authors cannot compare to previous tests. They should test different coating methods. For example, different TiO2 concentrations, or different coating materials.
  7. The language and grammar need to be improved throughout the manuscript

Eg. Be consistent in the manuscript. If you use 15wt%, please change all 15% by mass to 15wt%.

Eg. In the abstract “To analyze the quality and adhesion of coatings there was applied a scratch test”

Eg. Introduction line 31: “Due to the unique combination of thermal and mechanical properties, of particular technical importance are TiO2 and Al2O3”

Author Response

Thank you very much for the valuable comments. 

  1. Fixed abstract with the brief conclusion
  2. In the introduction:

- information on details of HVOF methods has been reduced. Advantages of this process have been indicated.

- information on Al2O3-TiO2 coatings thermally applied to the surface of castings in the energy industry with emphasis on their applications as coatings enhance the life of these materials by making them resistant to erosion and corrosion was introduced. It is worth noting that a much thicker coating thickness can be achieved by thermal spraying, which is another requisite for such applications.

  1. Information is given on the introduction of various contents of TiO2 into the Al2O3 coating. Indicating that the addition of TiO2 improves the brittlness, while retaning high hardness, a mixture with a certain small amount of TiO2 is favored i.e. Al2O3-13 t.% TiO2. It is also worth emphasizing that hardness and porosity changed with TiO2 content in Al2O3-TiO2 coatings.
  2. The choice of the Al2O3-15 wt.% TiO2 composition resulted from previous expert studies on the coating of iron and Al alloys of industrial fan blades in flue gas desulphurization installations. The possibility of replacing hardfaced coatings, which have a better service life, but are thicker and increase weight, was investigated moving parts of machines.
  1. The article mainly refers to the anti-wear properties of oxide coatings, taking into account parameters such as hardness and the introduction of an interlayer to improve abrasion resistance.
  2. Research purpose has been improved:

- characterized system in detail and investigated the impact of microstructure, phase composition, physical and mechanical properties on wear properties, with special focus on NiAl impact

  1. The English language was improved in the indicated examples and new references were introduced

Moreover, introduced a record: 15 wt.%TiO2.  

Please find in attachment file changes in red.

Reviewer 3 Report

- The preparation of the samples for the experiment is correct

- The laboratory devices have right performance and are suitable for the experiment

- The experiment is correct

- The obtained results provide  the certainty of a quality scientific paper.

- The results  of the experiments and their interpretation offer novelty and scientific value

Author Response

Thank you very much for review. 

Round 2

Reviewer 2 Report

The authors answered all my questions and improved the manuscript based on my suggestions. I don't have any more comments in the second review process.